# UDP-Glucose: A Cereblon-Dependent Glucokinase Protein Degrader

**DOI:** 10.3390/ijms23169094

**Published:** 2022-08-13

**Authors:** Jaeyong Cho, Atsushi Miyagawa, Kazuki Yamaguchi, Wakana Abe, Yoji Tsugawa, Hatsuo Yamamura, Takeshi Imai

**Affiliations:** 1Department of Chemical Biology, National Center for Geriatrics and Gerontology, Obu 474-8511, Aichi, Japan; 2Department of Life Science and Applied Chemistry, Nagoya Institute of Technology, Gokiso-cho, Showa-ku, Nagoya 466-8555, Aichi, Japan

**Keywords:** diabetes, insulin, MODY, UDP-glucose, glucokinase, glucose, protein degrader, PROTACs

## Abstract

We previously reported that glucokinase is ubiquitinated and degraded by cereblon with an unknown endogenous glucokinase protein degrader. Here, we show that UDP-glucose is a glucokinase protein degrader. We identified that both glucose and UDP-glucose bind to glucokinase and that both uridine and UDP-glucose bind to cereblon in a similar way to thalidomide. From these results, UDP-glucose was identified as a molecular glue between cereblon and glucokinase. Glucokinase produces glucose-6-phosphate in the pancreas and liver. Especially in β-cells, glucokinase is the main target of glucose for glucose-induced insulin secretion. UDP-glucose administration ubiquitinated and degraded glucokinase, lowered glucose-6-phosphate production, and then reduced insulin secretion in β-cell lines and mice. Maturity-onset diabetes of the young type 2 (MODY2) glucokinase^E256K^ mutant protein was resistant to UDP-glucose induced ubiquitination and degradation. Taken together, glucokinase ubiquitination and degradation signaling might be impaired in MODY2 patients.

## 1. Introduction

The main target of glucose in glucose-induced insulin secretion (GSIS) is glucokinase (GCK [1]). Glucokinase phosphorylates glucose and produces glucose-6-phosphate (G6P). Elevated G6P leads to insulin secretion stimulation. Single glucokinase variants were found in maturity-onset diabetes in young type 2 (MODY2) patients [1,2,3]. Dominant variants of glucokinase in MODY2 patients displayed glucose-intolerant hypoinsulinemia [2]. Our previous study showed that glucokinase is one of the arginine targets for insulin secretion [1], and we found arginine- and glucose-intolerant hypoinsulinemia in a MODY2 patient with the glucokinase^E442*/WT^ variant [1]. Both arginine and glucose bind to glucokinase similarly [1]. Additionally, point mutagenesis analysis showed that the E256, E442, and E443 residues are indispensable for binding to arginine [1]. The MODY2 variant glucokinase^E256K^ has less binding activity to arginine [1].

Protein degraders and proteolysis-targeting chimeras (PROTACs) are new types of drugs that degrade (undruggable) target proteins with the E3 ubiquitin ligases cereblon (CRBN), von Hippel–Lindau (VHL), murine double minute 2 (MDM2), and cellular inhibitor of apoptosis protein 1 (cIAP1) [4]. Among these E3 ubiquitin ligases, cereblon is the only chemically inducible E3 ubiquitin ligase [4]. Cereblon is a component of the cullin ring ubiquitin ligase 4 (CRL4) complex, and the CRL4 complex consists of the cullin 4A/B (CUL4)-DDB1-cereblon (CRL4^CRBN^ [5,6,7]). In the cereblon-dependent protein degrader or the cereblon-based molecular glue, there are at least three moieties, cereblon-binding moiety, substrate protein-binding moiety, and linker [4]. The cereblon-binding moiety is a glutarimide or succinimide, such as IMiDs (thalidomide, lenalidomide, and pomalidomide) and uridine [4]. Ito et al. (2015) showed that the cereblon^YWAA^ mutant has no thalidomide-binding activity [8]. This site is indispensable for thalidomide-binding residues [1,8,9]. On the other hand, substrate protein (glucokinase)-binding moiety might be glucose derivatives, such as G6P and UDP-glucose [10]. Non-glucose hexoses, such as galactose and N-acetyl-glucosamine, do not have an affinity for glucokinase. Glucokinase is a member of the hexokinases, which produce G6P and is the first and rate-limiting step of glycolysis [11]. Ubiquitinated and degraded glucokinase might lead to the suppression of glycolysis.

Here, we identified UDP-glucose as an endogenous glucokinase protein degrader. UDP-glucose administration to β-cells degrades glucokinase and reduces G6P production and insulin secretion. In hepatocytes, UDP-glucose reduces G6P production, glycolysis, and cell growth, although UDP-glucose does not change hepatocyte function.

## 2. Results

### 2.1. Glucose Inhibits Glucokinase Ubiquitination and Degradation

We previously reported that arginine behaves as a glucokinase-binding antagonist for cereblon [CRL4^CRBN^ (CUL4–DDB1–RBX1–cereblon)]-dependent glucokinase^WT^ protein ubiquitination and degradation [1,5,6,7]. Arginine depletion in the medium triggers glucokinase^WT^ degradation, but no apparent degradation was observed in the glucokinase^E256K^-mutated protein [1]. In addition, we also show that glucokinase^E256K^ protein has less binding activity to arginine [1]. We conclude that arginine binds to E256 residue and behaves as a cereblon-dependent degradation inhibitor [1].

Glucokinase^E256K^ has much less binding activity to glucose than glucokinase^WT^ in vitro (Figure 1a). Glucokinase^WT^ bound to glucose, although it did not bind to UTP nor UDP in vitro (Figure 1b). These in vitro data indicated that glucose might behave as a glucokinase-binding antagonist for ubiquitination and degradation by cereblon, and glucokinase^E256K^-mutated protein was resistant to ubiquitination similar to arginine (Appendix A and [1]). A possible molecular model is shown in Appendix A. The E256 residue was involved in arginine and glucose binding, and glucose had no binding activity for cereblon (Figure 2c).

Glucose derivatives, such as galactose and N-acetyl-glucosamine (NAG), do not bind to glucokinase in vitro (Figure 1c). On the other hand, glucose-containing chemicals, such as glucose and UDP-glucose, bind to glucokinase in vitro (Figure 1d). Unknown glucokinase protein degrader X may contain glucose or arginine as its glucokinase-binding moiety ([4] and Figure 1f). MEIS2 has been identified as an unknown ligand-dependent substrate protein [5]. MEIS2 does not bind to glucose, UDP, or UTP in vitro (Figure 1e); thus, glucokinase protein degrader X might not be a MEIS2 protein degrader.

### 2.2. Uridine Binds to Cereblon

Next, we further analyzed endogenous cereblon-binding chemicals. The first identified cereblon-binding chemical is thalidomide [8]. Immunomodulatory drugs (IMiDs; thalidomide, lenalidomide, and pomalidomide), uridine, and UDP-glucose contain glutarimide moieties (Figure 2a, and [12]). The FLAG-tagged cereblon^YWAA^ ([1] and YWAA-F) mutant has no binding activity to thalidomide ([1,8,9] and Figure 2b). FLAG-tagged cereblon^WT^ ([1], and CRBN-F) binds to UDP and UTP but does not bind to glucose in vitro (Figure 2c); nevertheless, the cereblon^YWAA^ mutant has no binding activity to nether, glucose, UDP, or UTP (Figure 2d). UDP and UTP binding to cereblon is similar to thalidomide binding to cereblon. All the cereblon-binding chemicals contain glutarimide moieties (Figure 2a,e).

### 2.3. UDP-Glucose Induces Cereblon-Dependent Glucokinase Ubiquitination and Degradation

Previous results showed that glucose binds to glucokinase (Figure 1), and uridine and UDP-glucose bind to cereblon^WT^, although the cereblon^YWAA^ mutant does not bind uridine or UDP-glucose (Figure 2). These data indicated that UDP-glucose might be the molecular glue between cereblon and glucokinase and might be the endogenous glucokinase protein degrader X ([1] and Figure 3h).

To confirm this hypothesis, UDP-glucose- and cereblon-dependent ubiquitination of glucokinase was analyzed (Figure 3 and Appendix A). For glucokinase-high expression in the pancreas and liver [1], HA-tagged glucokinase, FLAG-tagged cereblon, and Myc-tagged ubiquitin (Ubi-Myc) expression vectors were transfected into HEK293 cells. Then, UDP-glucose was administered, and HA-tagged glucokinase protein was analyzed (Figure 3a and Appendix A). The cells with the Myc-tagged ubiquitin expression vector displayed significantly (*p* < 0.0001) higher HA-tagged glucokinase ubiquitination than those without the Myc-tagged ubiquitin vector with/without UDP-glucose administration (comparison of empty box and filled box in Figure 3a), indicating that glucokinase was ubiquitinated protein. Moreover, UDP-glucose administration induced significantly glucokinase ubiquitination (*p* = 4.30802 × 10^−7^) in a dose-dependent manner. Administered UDP-glucose induced cereblon-dependent exogenous glucokinase ubiquitination.

UDP-glucose was administered to pancreatic NIT-1 cells, in which endogenous glucokinase, ubiquitin, and cereblon were expressed. Endogenous glucokinase protein amount was analyzed and degraded in a dose-dependent manner (Figure 3b,c). The degradation occurred from 3 to 72 h after administration (Appendix A), although transcription of glucokinase was unchanged in this period (Appendix A).

We analyzed the intracellular UDP-glucose concentration in several cell lines (Figure 3d). The highest intracellular concentration was observed in HEK293 cells, and one of the lowest concentrations was observed in multiple myeloma NCI-H929-1a cells (Figure 3d). UDP-glucose was administered to both cell lines, and glucokinase protein levels were analyzed by WB (Figure 3e,f). In HEK293 cells, 10 mM UDP-glucose administration reduced glucokinase (Figure 3e). However, 1 mM UDP-glucose was enough to reduce glucokinase protein in NCI-H929-1a cells (Figure 3f). These data indicated that both endogenous and administered UDP-glucose degraded glucokinase protein.

Finally, the glucokinase degradation activity of other UDP-monosaccharides, such as UDP-NAG and UDP-galactose, and IMiDs (thalidomide, lenalidomide, pomalidomide) was analyzed (Figure 3g). Among these compounds, only UDP-glucose administration induced glucokinase degradation among IMiDs and UDP-monosaccharide. UDP-glucose is the glucokinase protein degrader and molecular glue between glucokinase and cereblon.

### 2.4. UDP-Glucose Induces Glucokinase Degradation and Reduces Insulin Secretion from 3 h to 24 h

In Figure 3, we demonstrated that UDP-glucose was the first identified glucokinase degrader. Glucokinase is the rate-limiting factor for GSIS. Glucokinase phosphorylates glucose and produces G6P [1]. Therefore, we studied the UDP-glucose effect on insulin secretion after glucokinase degradation in β-cells (Figure 4). The amount of G6P in NIT-1 cells was significantly reduced by UDP-glucose administration (Figure 4a). UDP-glucose behaves as the molecular glue between glucokinase and cereblon in HEK293 cells (Figure 4b,c) as well as in vitro (Figure 4d). Therefore, UDP-glucose administration reduces insulin secretion in NIT-1 cells (Figure 4e) and mice (Figure 4f). In addition, UDP-glucose induced glucose concentration due to reduced insulin secretion (Figure 4g).

UDP-glucose administration in endogenonus glucokinase-expressing Hep G2 cells (Appendix A) and exogenous HA-tagged glucokinase-expressing HEK293 cells (Appendix A) reduced G6P production in a cereblon-dependent manner (Appendix A). Other UDP-monosaccharides, such as UDP-galactose and UDP-NAG, which have no glucokinase ubiquitination and degradation activity (Figure 3g), also exhibited no G6P reduction (Appendix A) or reduced insulin secretion (Appendix A).

UDP-glucose pyrophosphorylase (UGP) is the enzyme of UDP-glucose synthesis.
Glucose-1-phosphate + UTP ⇌ UDP-glucose + pyrophosphate

UGP protein was knocked down (KD) in NIT-1 cells (Figure 4h). UDP-glucose was administered to these cells, and glucokinase protein degradation was analyzed (Figure 4i,j). Apparent glucokinase protein degradation was not observed with UGP-KD, but glucokinase protein degradation and G6P reduction by UDP-glucose administration were similarly observed in control and UGP-KD NIT-1 cells (Figure 4i,k). UGP-KD induced glucokinase protein (Figure 4i) and G6P concentration significantly (*p* = 0.00422 and Figure 4k), indicating that endogenous UDP-glucose produced by UGP induced glucokinase protein degradation and reduced intracellular G6P production.

Taken together, UDP-glucose administration in pancreatic β-cells first induces an interaction between cereblon and glucokinase as the molecular glue and then induces ubiquitination and degradation of glucokinase. Degraded glucokinase also reduced G6P production and insulin secretion (Figure 4).

### 2.5. Uridine-Glucose Degrades Glucokinase and Reduces Insulin Secretion

We identified UDP-glucose as being an endogenous glucokinase degrader (Figure 3 and Figure 4). Glucose binds to glucokinase (Figure 1), and the glutarimide moiety from uridine binds to Y384 and W386 residues of cereblon (Figure 2). The molecular glue contains three parts: an E3 ubiquitin ligase (cereblon)-binding moiety, a substrate protein (glucokinase)-binding moiety, and a linker between the two [4]. We tested uridine-glucose (U-glucose, Figure 5a) without a diphosphate linker in UDP-glucose to analyze the effect of the linker. U-glucose and UDP-glucose were administered to NIT-1 cells, and intracellular glucokinase protein and G6P and secreted insulin were analyzed (Figure 5b–d). U-glucose and UDP-glucose similarly degraded glucokinase protein (Figure 5b). The substrates glucokinase protein (Figure 5b), G6P (Figure 5c), and insulin (Figure 5d) also decreased in a dose-dependent manner.

### 2.6. UDP-Glucose Reduced G6P Production in Hepatocytes 

Glucokinase is expressed at its highest levels in pancreatic β-cells and the liver [1]. Therefore, UDP-glucose was administered to Hep G2 cells (Figure 6). At first, UDP-glucose administration reduced the intracellular glucose-6-phosphate (G6P) concentration (Figure 6a). G6P is the first step of glycolysis, and 2-deoxy-d-glucose (2DG) is the repressor of glycolysis [11]. Cell proliferation rate was analyzed with UDP-glucose (Figure 6b). Hep G2 proliferation was significantly reduced by UDP-glucose administration (Figure 6b). However, UDP-glucose administration did not change in terms of hepatocyte function during the bromsulphalein test (Figure 6c).

## 3. Discussion

Thalidomide was the first identified cereblon-dependent protein degrader [8]. Uridine was also the first identified endogenous cereblon-dependent protein degrader, but target/substrate protein(s) are still unknown [13]. Here, we identified UDP-glucose as an endogenous cereblon-dependent glucokinase degrader. Endogenous UDP-glucose-dependent protein degradation is different from exogenous thalidomide. Thalidomide administration triggers protein degradation, although endogenous UDP-glucose always degrades glucokinase protein in NIT-1 (Figure 4b–k) and HEK293 (Appendix A) cells. The UDP-glucose pyrophosphorylase (UGP) is the only enzyme that catalyzes the synthesis of UDP-glucose from glucose-1-phosphate and UTP. The UGP was knocked down to decrease endogenous UDP-glucose, and higher glucokinase protein and G6P concentrations were observed in the UGP-KD cells (Figure 4h–k).

Reduced fasting glucose levels and arginine concentration lead to decreased insulin secretion [1]. This phenomenon was reflected in a glucose- and arginine-depleted medium experiment (Appendix A and [1]). Other UDP-monosaccharides—UDP-galactose and UDP-NAG—might be competitive with endogenous UDP-glucose and induced insulin secretion (Appendix A). Nutrition chemicals also regulated UDP-glucose- and cereblon-dependent glucokinase protein degradation. Taken together, endogenous UDP-glucose degraded cereblon-dependent glucokinase protein.

On the other hand, administered UDP-glucose also degraded cereblon-dependent glucokinase protein degradation in NIT-1 cells (Figure 4a,e,h–k), Hep G2 cells (Appendix A and Figure 6), HEK293 cells (Appendix A), and mice (Figure 4f,g). Similarly, uridine-glucose administration also degraded glucokinase and reduced G6P and insulin secretion (Figure 5). There are two points to consider for UDP-glucose administration. One is administration amount. Endogenous intracellular UDP-glucose concentration is affected by the amount administered (Figure 3d–f). The other is administration time. Nutrition state affected the degradation (Appendix A and [1]). Taken together endogenous and administered UDP-glucose induced cereblon-dependent glucokinase degradation.

## 4. Materials and Methods

### 4.1. Antibodies and Cell Culture

The following antibodies were purchased: β-actin (sc-47778, Santa Cruz Biotechnology, Santa Cruz, CA, USA), FLAG (F-1804, Sigma, St. Louis, MO, USA), and UDP-glucose pyrophosphorylase (UGP, sc-377089 Santa Cruz). Mouse pancreas-derived NIT-1 cells (CRL-2055TM purchased from ATCC, Manassas, VA, USA) were maintained as described previously [1,14,15,16,17,18,19,20,21]. Briefly, the cells were plated at a density of 1.5–3.0 × 10^6^ cells/60 mm dish with replacement of F-12K medium (Kaighn’s modification of Ham’s F-12 medium) plus 10% fetal calf serum (FCS) after 48 h of culture. Human embryo kidney 293T (HEK293T) cells and human hepatocellular carcinoma (Hep G2) cells were maintained with DMEM + 10% FCS [20,21,22,23]. Cereblon KO cells were kindly provided by Tokyo Medical University [8]. UGP knockdown siRNA (sc-154894) was purchased from Santa Cruz.

### 4.2. Analysis of Insulin Secretion from Cells

Insulin secretion was determined using a commercial enzyme-linked immunosorbent assay (ELISA) kit (Shibayagi, Gunma, Japan) [15,16,17,18,19,20,21,22] as described previously. Briefly, the culture medium of the NIT-1 cells was replaced with UDP-glucose-containing F12K + 10% fetal calf serum (FCS) for the indicated time. Subsequently, the medium was collected for insulin measurement.

### 4.3. Immunoprecipitation and Western Blot

Immunoprecipitation and Western blot analysis were performed as described previously with minor modifications [14,15,16,17,18,19,20,21]. NIT-1 cells transfected with the pCDNA-insulin-Myc expression vector were cultivated in arginine-free medium for 30 min before arginine was added. Cells were lysed in buffer containing 20 mM HEPES-NaOH (pH 7.9), 1 mM MgCl_2_, 0.2 mM CaCl_2_, 100 mM KCl, 0.2 mM EDTA, 10% glycerol, 0.1% Nonidet P-40, 1 mM dithiothreitol, 0.2 mM phenylmethylsulfonyl fluoride (Nacalai Tesque, Kyoto, Japan), and 3% n-octyl-β-d-glucoside (DOJINDO, Tokyo, Japan). After incubation on ice for 30 min, lysates were centrifuged at 12,000× *g* for 15 min at 4 °C and dialyzed for 3 h at 4 °C to lysis buffer without n-octyl-β-d-glucoside. After dialysis, the supernatant was incubated with the indicated antibody for 120 min at 4 °C. The samples were then incubated with Protein G Sepharose 4 Fast Flow (GE Healthcare, Chicago, IL, USA). After an additional washing, the precipitates were heat-denatured in SDS-sample buffer. For immunoblot analysis, proteins were separated by SDS/PAGE and transferred to a PVDF membrane for Western blot analysis. The membranes were blocked with 5% nonfat milk and Tris-buffered saline with Tween 0.1%, incubated overnight with a mixture of primary antibody (c-Myc [1:1000]) and Can Get Signal solution (TOYOBO, Osaka, Japan) at 4 °C, washed, incubated with the secondary antibody for 60 min at room temperature, and then washed again. Immune complexes were detected using Immunostar LD (Fuji-film, Tokyo, Japan) substrate. Signals were quantified with the LAS 4000 imaging system (GE Healthcare). ImageJ was used for densitometry of scanned membranes [14,15,16,17,18,19,20,21].

### 4.4. Chemical Detection

Intracellular G6P and UDP-glucose concentration was determined in two ways [1]. One is a commercial kit (Glucose-6-Phosphate Fluorometric Assay Kit No. 700750, Cayman chemical, Ann Arbor, MI, USA [1]). Another is metabolome analysis [1,22,23].

### 4.5. Chemical Synthesis

#### 4.5.1. General

^1^H and ^13^C NMR spectra were recorded in CDCl_3_-d and D_2_O using a Bruker AVANCE 400 Plus Nanobay instrument (400 and 101 MHz) and a Bruker AVANCE 500 instrument fitted with a cryoprobe (500 and 126 MHz), respectively. Chemical shifts (δ) are given in ppm and referenced to tetramethylsilane (0.00 ppm) or the internal solvent signal used as an internal standard. Assignments in the NMR spectra were made by first-order analysis of the spectra and were supported by ^1^H−^1^H COSY and ^1^H−^13^C HMQC correlation results. Matrix-assisted laser desorption ionization time-of-flight high-resolution mass spectrometry (MALDI-TOF HRMS) spectra were recorded on a Jeol JMS-S3000 using 2,5-dihydroxylbenzoic acid as the matrix. Unless otherwise stated, all commercially available solvents and reagents were purchased from FUJIFILM Wako Pure Chemical Corporation and Tokyo Chemical Industry Company Limited without further purification.

#### 4.5.2. 2-Bromoethyl 2,3,4,6-Tetra-O-acetyl-β-d-glucopyranoside (**2**)

To a solution of compound 1 (3.00 g, 6.09 mmol) and 2-bromoethanol (743 μL, 10.4 mmol) in dichloromethane (28.4 mL), molecular sieves 3 Å powder (3.00 g) was added and cooled at −40 °C. To the mixture was added boron trifluoride-ethyl ether complex (229 μL, 1.82 mmol) diluted in dichloromethane (2.1 mL) and stirred at −40 °C for 30 min. The solution was diluted with dichloromethane and filtered through celite. The filtrate was washed with aqueous sodium hydrogen carbonate and brine, dried over anhydrous sodium sulfate, filtered, and evaporated. The residue was purified by silica gel chromatography with 8:1 to 4:1 (*v*/*v*) hexane-ethyl acetate to give 2 (1.84 g, 66%): 1H NMR (CDCl3, 400 MHz) δ 5.22 (t, 1 H, J2,3 = 9.5 Hz, H-3), 5.09 (t, 1 H, J3,4 = 9.5 Hz, H-4), 5.02 (dd, 1 H, Hz, H-2), 4.58 (d, 1 H, J1,2 = 8.0 Hz, H-1), 4.58 (dd, 1 H, J = 4.8 Hz, J6a,6b = 12.3 Hz, H-6a), 4.20–4.13 (m, 2 H, H-6b, -OCH2-), 3.85–3.79 (m, 1 H, -OCH2-), 3.74–3.69 (m, 1 H, H-5), 3.49–3.45 (m, 2 H, -CH2Br), 2.10 (s, 3 H, Ac), 2.08 (s, 3 H, Ac), 2.03 (s, 3 H, Ac), 2.02 (s, 3 H, Ac); 13C{1H} NMR δ (CDCl3, 101 MHz): 170.6, 170.2, 169.4, 169.4, 101.0, 72.6, 71.9, 71.0, 69.8, 68.3, 61.8, 29.8, 20.7, 20.6, 20.5.

#### 4.5.3. 6-Bromohexyl 2,3,4,6-Tetra-O-acetyl-β-d-glucopyranoside (**3**)

To a solution of compound 1 (3.00 g, 6.09 mmol) and 6-bromoethanol (2.49 mL, 10.3 mmol) in dichloromethane (28.4 mL) were added molecular sieves of 4 Å powder (3.00 g) and cooled at −40 °C. To the mixture was added boron trifluoride-ethyl ether complex (229 μL, 1.82 mmol) diluted in dichloromethane (2.1 mL) and stirred at −40 °C for 30 min, at −20 °C for 30 min and then at 0 °C for 30 min. The solution was diluted with dichloromethane and filtered through celite. The filtrate was washed with aqueous sodium hydrogen carbonate and brine, dried over anhydrous sodium sulfate, filtered, and evaporated. The residue was purified by silica gel chromatography with 10:1 to 3:1 (*v*/*v*) hexane-ethyl acetate to give 3 (2.06 g, 66%): 1H NMR (CDCl3, 400 MHz) δ 5.21 (t, 1 H, J2,3 = 9.5 Hz, H-3), 5.09 (t, 1 H, J3,4 = 9.7 Hz, H-4), 4.99 (dd, 1 H, Hz, H-2), 4.49 (d, 1 H, J1,2 = 8.0 Hz, H-1), 4.27 (dd, 1 H, J = 4.7 Hz, J6a,6b = 12.3 Hz, H-6a), 4.14 (dd, 1 H, J5,6a = 2.4 Hz, H-6b), 3.91–3.85 (m, 1 H, -OCH2-), 3.72–3.67 (m, 1 H, H-5), 3.51–3.46 (m, 1 H, -OCH2-), 3.41 (t, 2 H, J = 6.8 Hz, -CH2Br), 2.11 (s, 3 H, Ac), 2.05 (s, 3 H, Ac), 2.03 (s, 3 H, Ac), 2.01 (s, 3 H, Ac), 1.89–1.82 (m, 2 H, -CH2-), 1.62–1.55 (m, 2 H, -CH2-), 1.46–1.32 (m, 4 H, -CH2-); 13C{1H} NMR (CDCl3,101 MHz) δ 170.7, 170.3, 169.4, 169.2, 100.8, 72.8, 71.7, 71.3, 69.9, 68.4, 62.0, 33.7, 32.6, 29.2, 27.7, 25.0, 20.7, 20.6, 20.6, 20.6.

#### 4.5.4. 2-S-Thioacetylethyl 2,3,4,6-Tetra-O-acetyl-β-d-glucopyranoside (**4**)

To a solution of compound 2 (500 mg, 1.10 mmol) and thioacetic acid (117 μL, 1.64 mmol) in N,N-dimethylformamide (3.67 mL) was added potassium carbonate (228 mg, 1.65 mmol) and stirred at room temperature for 12 h. The solution was diluted with ethyl acetate, washed with aqueous sodium hydrogen carbonate and brine, dried over anhydrous sodium sulfate, filtered, and evaporated. The residue was purified by silica gel chromatography with 4:1 (*v*/*v*) hexane-ethyl acetate to give 4 (492 mg, 99%): 1H NMR (CDCl3, 400 MHz) δ 5.21 (t, 1 H, J2,3 = 9.5 Hz, H-3), 5.09 (t, 1 H, J3,4 = 9.7 Hz, H-4), 4.99 (dd, 1 H, Hz, H-2), 4.53 (d, 1 H, J1,2 = 8.0 Hz, H-1), 4.26 (dd, 1 H, J = 4.8 Hz, J6a,6b = 12.3 Hz, H-6a), 4.14 (dd, 1 H, J = 2.4 Hz, J6a,6b = 12.3 Hz, H-6b), 4.01–3.95 (m, 1 H, -OCH2-), 3.73–3.68 (m, 1 H, H-5), 3.65–3.59 (m, 1 H, H-5, -OCH2-), 3.26–3.01 (m, 2 H, -CH2S), 2.34 (s, 3 H, SAc), 2.09 (s, 3 H, Ac), 2.08 (s, 3 H, Ac), 2.03 (s, 3 H, Ac), 2.01 (s, 3 H, Ac); 13C{1H} NMR (CDCl3, 101 MHz) δ 195.2, 170.6, 170.2, 169.4, 169.3, 100.8, 72.7, 71.9, 71.1, 68.6, 68.3, 61.8, 30.5, 28.8, 20.7, 20.7, 20.6, 20.5.

#### 4.5.5. 6-S-Thioacetylhexyl 2,3,4,6-Tetra-O-acetyl-β-d-glucopyranoside (**5**)

To a solution of compound 3 (700 mg, 1.37 mmol) and thioacetic acid (146 μL, 1.79 mmol) in N,N-dimethylformamide (4.57 mL) was added potassium carbonate (284 mg, 2.05 mmol) and stirred at room temperature for 17.5 h. The solution was diluted with ethyl acetate, washed with water and brine, dried over anhydrous sodium sulfate, filtered, and evaporated. The residue was purified by silica gel chromatography with 8:1 to 4:1 (*v*/*v*) hexane-ethyl acetate to give 5 (561 mg, 83%): 1H NMR (CDCl3, 400 MHz) δ 5.20 (t, 1 H, J2,3 = 9.5 Hz, H-3), 5.09 (t, 1 H, J3,4 = 9.7 Hz, H-4), 4.98 (dd, 1 H, Hz, H-2), 4.49 (d, 1 H, J1,2 = 8.0 Hz, H-1), 4.27 (dd, 1 H, J = 4.7 Hz, J6a,6b = 12.3 Hz, H-6a), 4.14 (dd, 1 H, J = 2.4 Hz, J6a,6b = 12.3 Hz, H-6b), 3.90–3.84 (m, 1 H, -OCH2-), 3.71–3.67 (m, 1 H, H-5), 3.50–3.44 (m, 1 H, H-5, -OCH2-), 2.85 (t, 2 H, J = 7.3 Hz -CH2S), 2.33 (s, 3 H, SAc), 2.09 (s, 3 H, Ac), 2.04 (s, 3 H, Ac), 2.03 (s, 3 H, Ac), 2.01 (s, 3 H, Ac), 1.59–1.52 (m, 4 H, -CH2-), 1.37–1.32 (m, 4 H, -CH2-); 13C{1H} NMR (CDCl3, 101 MHz) δ 196.0, 170.7, 170.3, 169.4, 169.3, 100.8, 72.9, 71.8, 71.3, 69.0, 68.5, 62.0, 30.6, 29.4, 29.2, 29.0, 28.4, 25.3, 20.8, 20.7, 20.6, 20.6.

#### 4.5.6. 2-S-Ethyl-β-d-glucopyranose S-Linked Maleimide (**6**)

A solution of 4 (50.0 mg, 111 μmol) in methanol (950 μL) was added to 28% NaOMe-MeOH (50 μL) and then stirred at room temperature for 1 h. Diaion™ SK1B was added to the solution for quenching and filtration and evaporation. The residue was dissolved in methanol (2.22 mL), Et3N (22 μL), and maleimide (53.9 mg, 555 μmol) were added, and the mixture was stirred at room temperature for 1 h. The solution was concentrated, and the residue was purified by silica gel chromatography with 50:1 to 5:1 (*v*/*v*) dichloromethane-methanol to give 6 (30.3 mg, 81%): 1H NMR (D2O, 400 MHz) δ 4.30 (d, 1 H, J1,2 = 8.0 Hz, H-1), 3.98–3.90 (m, 2 H, -CH_2_SCH-, -OCH_2_-), 3.75–3.72 (m, 2 H, H-6, -OCH_2_-), 3.53 (dd, 1 H, J5,6a = 6.0 Hz, J6a,6b = 12.1 Hz, H-6), 3.31–3.25 (m, 2 H, H-3, H-5), 3.21–3.07 (m, 3 H, H-2, H-4, -SCHCH2-), 2.90–2.86 (m, 1 H, -CH_2_S-), 2.80–2.75 (m, 1 H, -CH2S-), 2.63–2.58 (m, 1 H, -SCHCH2-); 13C{1H} NMR (CDCl3, 126 MHz) δ 180.9, 179.7, 102.3, 75.8, 75.5, 72.9, 69.4, 69.0, 60.6, 41.5, 41.4, 37.2, 30.3; MALDI-TOF HRMS m/z: [M + Na]+: Calcd for C12H19N1O8S1 + Na+: 360.0724; Found: 360.62. [24]

#### 4.5.7. 6-S-Hexyl-β-d-glucopyranose S-Linked Maleimide (**7**)

A solution of 5 (50.0 mg, 101 μmol) in methanol (950 μL) was added to 28% NaOMe-MeOH (50 μL) and then stirred at room temperature for 2.5 h. Diaion™ SK1B was added to the solution for quenching, filtration, and evaporation. The residue was dissolved in methanol (2.02 mL), Et3N (20 μL), and maleimide (49.0 mg, 505 μmol) were added, and the mixture was stirred at room temperature for 1 h. The solution was concentrated, and the residue was purified by reverse-phase C18 silica gel chromatography with 1:0 to 4:1 (*v*/*v*) H2O-CH3CN to give 6 (27.5 mg, 72%): 1H NMR (D2O, 500 MHz) δ 4.32 (d, 1 H, J1,2 = 8.0 Hz, H-1), 3.98–3.93–3.90 (m, 1 H, -CH2SCH-), 3.82–3.78 (m, 2 H, H-6a, -OCH2-), 3.61–3.52 (m, 2 H, H-6b, -OCH2-), 3.38–3.30 (m, 2 H, H-3, H-5), 3.27–3.10 (m, 3 H, H-2, H-4, -SCHCH2-), 2.67–2.57 (m, 3 H, -SCHCH2-, -CH2S-), 1.52–1.47 (m, 4 H, -CH2-), 1.32–1.24 (m, 4 H, -CH2-); 13C{1H} NMR (CDCl3, 126 MHz) δ 181.2, 179.9, 102.0, 75.8, 75.7, 73.0, 70.4, 69.5, 60.6, 41.4, 37.2, 30.0, 28.4, 28.1, 27.4, 24.4; MALDI-TOF HRMS m/z: [M + Na]+: Calcd for C16H27N1O8S1 + Na+: 416.1350; Found: 415.1309 [24].

### 4.6. Statistical Analysis

The values are reported as the means ± standard error (n = 6) with six individual data points. Statistical significance (single-sided Student’s *t* test) is indicated in the figure legends as follows: * *p* < 0.05. n = 6 for statistical analysis [15,16,17,18,19,20,21,22]. For reproducibility of key experiments, we performed a total number of experiments exceeding five times, including experiments for setting conditions with similar results. Additionally, we employed multiple approaches to confirm one result.

### 4.7. Ethical Approval

The study is reported in accordance with ARRIVE guidelines. All mouse experiments were performed in accordance with the ethical guidelines for animal care of the National Center for Geriatrics and Gerontology, and the experimental protocols were approved by the Animal Care Committee of the National Center for Geriatrics and Gerontology [20,21].

## 5. Conclusions

We previously reported that glucokinase is ubiquitinated and degraded by cereblon with the unknown glucokinase degrader X [1]. Here, we identified that glucokinase protein degrader X is UDP-glucose. UDP-glucose acted as the molecular glue connected with cereblon and glucokinase. The glutarimide moiety located in uridine binds to Y384 and W386 residues of cereblon, and the glucose moiety binds to glucokinase. The shorter linker chemical uridine-glucose acted as a glucokinase degrader similar to UDP-glucose. Glucokinase was highly expressed in the pancreas and liver. Both endogenous and administered UDP-glucose degraded cereblon-dependent glucokinase protein and reduced G6P and insulin secretion in the pancreas. UDP-glucose administration to hepatocytes revealed reduced hepatocyte proliferation but did not change hepatocyte function.

## Figures and Tables

**Figure 1 ijms-23-09094-f001:**
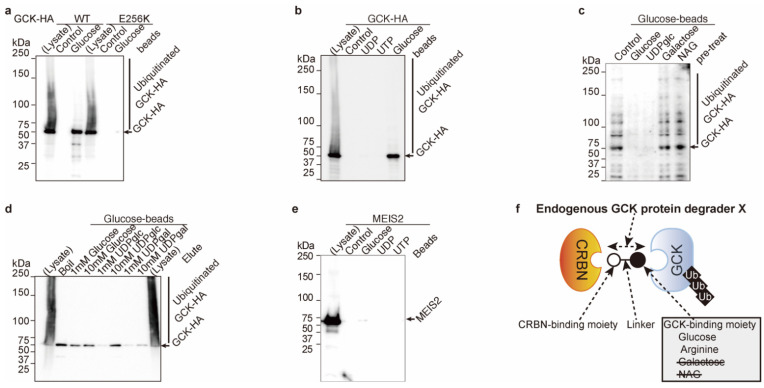
Glucose inhibits glucokinase (GCK) ubiquitination and degradation. (**a**) The E256 residue of glucokinase is involved in glucose binding. Glucokinase^WT^ and glucokinase^E256K^ proteins were mixed with control or glucose-immobilized beads, and the binding fraction was eluted by boiling. Glucokinase proteins were detected by WB. (**b**) Glucokinase does bind to glucose but does not bind to UDP or UTP. Glucokinase protein was mixed with control-, UDP-, UTP- or glucose-immobilized beads, and bead-binding glucokinase protein was dissociated by boiling from beads. Beads binding glucokinase were detected by WB. (**c**) Glucokinase binds to glucose and UDP-glucose and does not bind to galactose and N-acetyl glucosamine (NAG). Glucokinase proteins were pretreated with control, glucose, UDP-glucose, galactose or N-acetyl glucosamine (NAG) and then mixed with glucose beads. Beads binding glucokinase was eluted by boiling. Glucose beads binding glucokinase were detected by WB. (**d**) Glucokinase binds to glucose, and UDP-glucose does not bind to UDP-galactose. Glucose beads binding glucokinase proteins were eluted by glucose, UDP-glucose, or UDP-galactose. Eluted glucokinase was detected in glucose and UDP-glucose elution but not in UDP-galactose elution. (**e**) Cereblon (CRBN) substrate protein MEIS2 has no binding activity to glucose, UDP, and UTP. MEIS2 protein was mixed with control, glucose, UTP, and UDP-immobilized beads, and the beads were washed and boiled. Bead-bound MEIS2 was detected by WB. (**f**) Graphic summary Glucokinase-binding moiety of endogenous glucokinase protein X was glucose and arginine.

**Figure 2 ijms-23-09094-f002:**
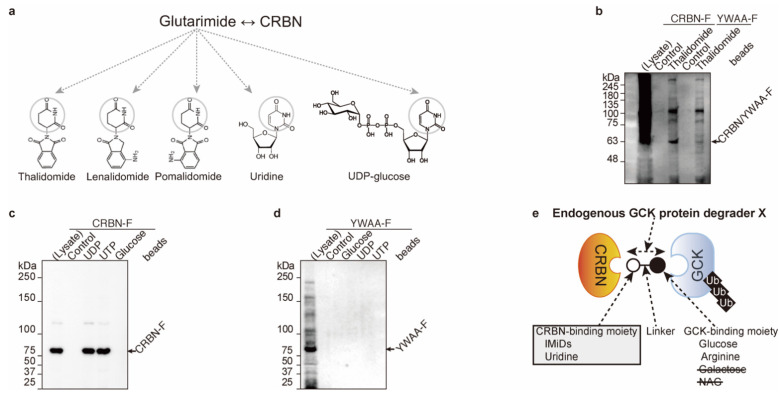
Uridine binds to cereblon (CRBN). (**a**) Glutarimide moiety (gray circles) in the IMiDs (thalidomide, lenalidomide, and pomalidomide), uridine, and UDP-glucose. (**b**–**d**) Cereblon binds to uridine and thalidomide. CRBN^WT^ proteins were mixed with control, thalidomide-immobilized beads (**b**), UDP and UTP-immobilized beads (**c**), control, and non-thalidomide-binding mutant CRBN^YWAA^ was mixed with control, thalidomide (**b**), UDP, and UTP (**d**)-immobilized beads. CRBN^WT^ binds to UDP, UTP, and thalidomide through Y384 and W386 residues. (**e**) Graphic summary Cereblon-binding moiety of endogenous glucokinase protein X was uridine.

**Figure 3 ijms-23-09094-f003:**
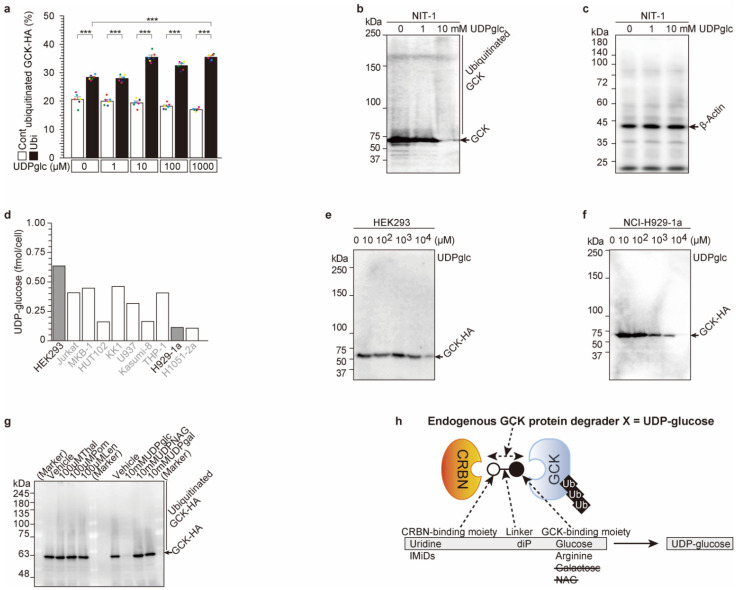
UDP-glucose induces cereblon-dependent glucokinase ubiquitination and degradation. (**a**) UDP-glucose induces glucokinase ubiquitination in a dose-dependent manner. The HEK 293 cells were transfected with HA-tagged glucokinase (GCK-HA), FLAG-tagged cereblon (CRBN-F), and Myc-tagged ubiquitin (ubi-Myc) expression vectors for one day, and then UDP-glucose was added for 3 h. Intracellular GCK-HA was detected by WB (Appendix A). Ubiquitinated GCK-HA (per total GCK-HA, %) was calculated and plotted. The values are reported as the means ± standard error (n = 6) with six individual data points. ***; *p* < 0.0001. (**b**,**c**) NIT-1 cells were treated with UDP-glucose for 3 h, the cells were harvested, and endogenous glucokinase proteins were detected by WB (**b**). Control β-actin WB is (**c**). (**d**–**f**) Both endogenous and administrated UDP-glucose regulated GCK-HA protein degradation. The endogenous UDP-glucose concentration was measured in various cell lines (**d**). Ten mM UDP-glucose administration reduced GCK-HA protein degradation in HEK293 cells (**e**). One hundred μM UDP-glucose administration reduced GCK-HA protein degradation in NCI-H929-1a cell (**f**). (**g**) UDP-glucose induces glucokinase degradation among UDP-monosaccharides (glucose, galactose and NAG) and IMiDs. (**h**) Graphic summary; Endogenous glucokinase protein X was UDP-glucose.

**Figure 4 ijms-23-09094-f004:**
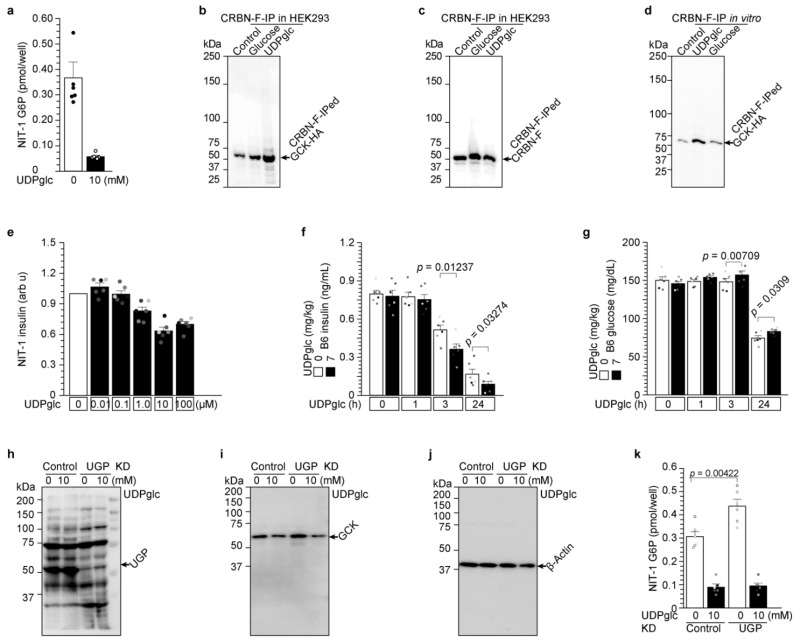
UDP-glucose induces glucokinase degradation and reduces insulin secretion from 3 h to 24 h. (**a**) UDP-glucose administration reduces glucose-6-phosphate (G6P) production in NIT-1 cells. UDP-glucose was administered to NIT-1 cells for 3 h, and then the cells were extracted. Intracellular G6P was analyzed using a commercial kit. (**b**,**c**) UDP-glucose induces the interaction of cereblon and glucokinase in HEK293 cells. Expression vectors of CRBN-F and GCK-HA were transfected into HEK 293 cells. These cells were treated with glucose and UDP-glucose for 3 h. The cells were harvested, and cell lysates were prepared. The precipitated fraction with anti-FLAG antibody was visualized with anti-HA (**b**) and anti-CRBN (**c**) antibody WB. (**d**) UDP-glucose induces the interaction of CRBN and glucokinase in vitro. HEK293 cells producing CRBN-F and GCK-HA proteins were mixed with control, glucose, and UDP-glucose in vitro for 3 h at 4 °C. The mixtures were precipitated with anti-FLAG antibody. IPed GCK-HA proteins were detected by Western blot with anti-HA antibody. (**e**) UDP-glucose reduces insulin secretion in NIT-1 cells. The values are reported as the means ± standard error (n = 6) with six individual data points. (**f**,**g**) Administration of the proinsulin protein degrader UDP-glucose to C57/BL6J mice significantly decreased circulating insulin (**f**) and increased glucose concentration (**g**). The values are reported as the means ± standard error (n = 6) with six individual data points. (**h**–**k**) Both endogenous and administered UDP-glucose degraded glucokinase using UDP-glucose pyrophosphorylase (UGP) knockdown in NIT1 cells. The values are reported as the means ± standard error (n = 6) with six individual data points (**k**).

**Figure 5 ijms-23-09094-f005:**
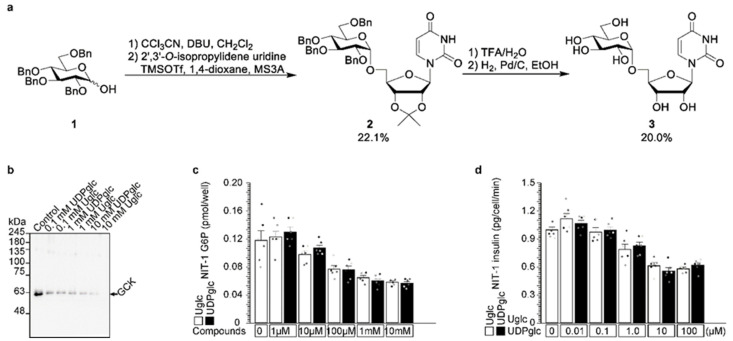
Uridine-glucose degraded glucokinase and reduced insulin secretion. (**a**) Uridine-glucose synthesis is described in the Materials and Methods. (**b**–**d**) Both uridine-glucose and UDP-glucose degraded glucokinase (**b**) and reduced G6P production (**c**) and insulin secretion (**d**) similarly. The values are reported as the means ± standard error (n = 6) with six individual data points (**c**,**d**).

**Figure 6 ijms-23-09094-f006:**
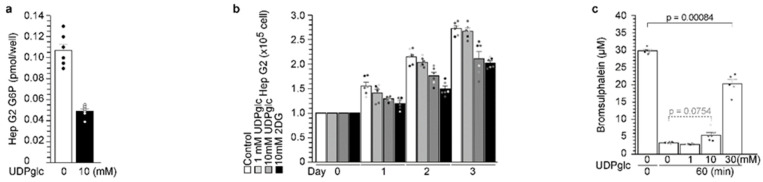
**UDP-glucose reduced G6P production in hepatocytes.** (**a**) UDP-glucose administration reduced the intracellular glucose-6-phosphate (G6P) concentration in Hep G2 cells. (**b**) UDP-glucose administration reduced the proliferation of Hep G2. (**c**) UDP-glucose administration did not change hepatocyte function. The values are reported as the means ± standard error (n = 6) with six individual data points (**a**–**c**).

## Data Availability

The data that support the findings of this study are available from the corresponding author upon reasonable request.

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
