# Peer review of "UDP-Glucose: A Cereblon-Dependent Glucokinase Protein Degrader"

_ijms, 2022, doi:10.3390/ijms23169094_

Round 1
Reviewer 1 Report
Summary
The submitted paper aims to show that UDP-glucose can act as glucokinase protein degrader. However, the clear, succinct conveyance of information is hindered by confusing, awkward, and grammatically incorrect writing in all the sections of the manuscript. In addition, most of the sentences are built on the elementary principle of noun, verb, and period, followed by an identically constructed simple sentence, which, to make matters worse, does not necessarily connect to the previous one. It looks like poorly conceived the art of writing succinctly, and thus the whole manuscript is hard to read and gives the impression of being sloppily written.
Abstract
UDP-glucose administration ubiquitinated and degraded glucokinase, reduces glucose-6-phosphate production, and then…
I believe that it was UDP-glucose that ubiquitinated and degraded glucokinase, and not its administration. Another issue is that the first part of this sentence is written in past simple, whereas the other part is in the present simple tense.
Introduction
the E256, E442 and E443 residues are indispensable for binding to arginine [1]. The MODY2 variant glucokinaseE256K has less binding activity to arginine [1].
Without any introduction, arginine is mentioned for the first time and the reason is obscure for a reader. On the other hand, more explanation is provided in the results section (2.1). I would consider briefly defining what arginine does with respect to the described protein.
Protein degraders and proteolysis-targeting chimeras (PROTACs) are new types of drugs that degrade (undruggable) target proteins with the E3 ubiquitin ligases cereblon (CRBN), von Hippel-Lindau (VHL), murine double minute 2 (MDM2), and cellular inhib-itor of apoptosis protein 1 (cIAP1) [4]. In particular, cereblon is a ligand-inducible E3 ubiquitin ligase, the cullin ring ubiquitin ligase 4 (CRL4) complex consisting of the cullin 4A/B (CUL4)-DDB1-cereblon (CRL4CRBN, [5-7]). In the protein degrader or molecular glue, there are at least three parts, cereblon-binding, substrate protein-binding, and linker…
This whole section is tough to follow. PROTAC, E3 ligases, CRL4 are all mentioned almost at once. It is difficult to distinguish between small molecules, proteins or protein domains. It needs rephrasing.
Cereblon binding moiety is a glutarimide or succinimide-containing chemical
Since cereblon is a protein, the authors should be referring to protein domains, and not moieties. Moreover, “chemical” from this sentence should be replaced with “moiety”.
In the protein degrader or molecular glue, there are at least three parts, cereblon-binding, substrate protein-binding, and linker [4].
Not all the protein degrades contain celebron-binding moiety. Please rewrite it for clarity.
Ubiquitinated and degraded glucokinase might lead to suppress glycolysis
Replace with “might lead to suppression of…”
Results
We previously reported that arginine behaves as a glucokinase-binding antagonist for cereblon [CRL4CRBN (CUL4–DDB1–RBX1–cereblon)]-dependent glucokinaseWT protein ubiquitination and degradation [1, 5-7]. Maturity-onset diabetes in young type 2 (MODY2) mutated glucokinaseE256K protein is resistant to this ubiquitination
It is difficult to understand how one sentence follows from another.
Unknown glucokinase degrader X may contain glucose or arginine for glucokinase binding, spacers/linkers, and cereblon-binding chemicals
All the proteins contain some kind of spacers or linkers that separate different protein domains. Is there anything special about these linkers in the degrader authors are referring to?
Immunomodulatory drugs (IMiDs; tha-lidomide, lenalidomide, and pomalidomide), uridine, and UDP-glucose contain glutarimide structures.
Replace “structures” with “moieties”. Alternatively, “Immunomodulatory drugs (IMiDs; tha-lidomide, lenalidomide, and pomalidomide), uridine, and UDP-glucose are all glutarimide derivatives.”
Glutarimide base
In chemistry, a base is a very specific term. Here, the authors are referring to “glutarimide moiety”.
(c), control, and. Non-thalidomide-binding mutant…
Sloppy writing, please correct it.
glucokinase, cereblon, and ubiquitin expression vectors were transfected into HEK293 cells. Then, UDP-glucose was administered, and glucokinase protein was analyzed (Fig. 3a). The cells with the ubiquitin expression vector displayed higher gluco-kinase ubiquitination than those without the ubiquitin vector
I believe many readers are here confused. Why do we observe increased ubiquitination? Are these overexpression vectors? If so, it is the overexpression itself that leads to a higher level of glucokinase ubiquitination? If glucokinase ubiquitination results in its degradation,
Figure 3
GCKHA suggests it is an acronym for protein. However, the caption explains that this is glucokinase tagged with HA. Therefore, GCKHA should be replaced with GCK-HA for clarity.
Another issue, what is cereblon-F? Where is an explanation?
Figure 3 (a) UDP-glucose induces glucokinase ubiquitination in a dose-dependent manner.
This is a Western-blot experiment, therefore it is hardly quantitative, especially since all the smears and almost all GCK-HA bands seem virtually the same.
Another issue: Where is a loading control for the cell line shown in (a)?
Figure 3 again:
endogenous glucokinase proteins
Why proteins? Are there different glucokinases?
Figure 3 again:
The cells were harvested and intracellular glucokinase-HA was detected by WB.
It seems rather obvious that the cells were harvested. Such statements should be removed from captions and moved to Materials and Methods.
Higher exogenous UDP-glucose is needed in higher UDP-glucose HEK293 cells
It seems a word is missing. Higher UDP-glucose level?
Lower exogenous UDP-glucose is needed
Same as above.
UDP-glucose was administered to pancreatic NIT-1 cells, in which endogenous glucokinase, ubiquitin, and cereblon were expressed.
We analyzed the intracellular UDP-glucose concentration in several cell lines…
Briefly mention how its concentration was determined. Which method was used?
UDP-glucose induced glucose concentration dew to reduce insulin secretion
What concentration? Lower/higher? Also several typos: it should be “due to reduced insulin”
Figure 4: and then the cells were extracted…
Probably, “harvested” is a better word.
We identified the endogenous glucokinase degrader UDP-glucose
Please rephrase it, for instance “We identified UDP-glucose as the endogenous glucokinase degrader.”
and glutarimide in uridine binds
Please rephrase it, for instance “glutarimide moiety from uridine”
(Fig. 2). molecular glue contains three parts:
A capital letter is required. In addition, I believe that many readers have forgotten by now what molecular glue is.
We explored uridine-glucose (U-glucose, Fig. 5a)
Please rephrase it, for instance “We tested uridine-glucose…”
Glucokinase expression is ß-cells and hepatocytes
Grammatically incorrect.
and 2DG is the repressor of glycolysis
2DG is not explained.
Discussion
This section must be re-written. It can hardly be called discussion, as there are only simple statements grouped together in this very short section. And again, any connection between the sentences is difficult to grasp and this is especially harmful for this section in particular.
Glucokinase is specifically expressed in pancreatic ß-cells and hepatocytes [1
What does it mean “specifically”? Is it only expressed in these cells? If so, why is it mentioned at the end of the manuscript?
Glucokinase degradation in pancreatic ß-cells reduces insulin secretion (Fig 4). How about
hepatocytes?
“How about” seems a bit awkward. Please rephrase it.
Hepatocyte function was also analyzed by the bromsulphalein assay (Fig 6c). No significant change was observed after UDP-glucose administra-tion. UDP-glucose reduces G6P and proliferation but has no apparent effect on hepatocyte function.
This looks like a statement from the result section, and not discussion. Please elaborate.
Author Response
Dear Reviewer 1,
Please see the attachment.
Thank you.
Takeshi Imai

Reviewer 2 Report
In this manuscript, Cho et al focused on identification of an endogenous degrader X that bridges the E3 ligase cereblon and glucokinase (GCK), a key enzyme in regulating glucose metabolism and implicated in MODY type of diabetes. The authors made a wise guess that glucose-UDP is the degrader X by showing that the glucose moiety binds to GCK and the UDP moiety binds to cereblon via the glutarimide structure conserved in uridine and other IMiD family of cereblon glue molecules. Overall, the findings and propositions are quite interesting. However, this reviewer has the following major concerns.
1. The language of this manuscript needs to be improved. Not only grammar wise, but also in terms of manuscript organization and proper introduction of prior relevant literature. The title, for example, is also not reflecting the main message of the work. Perhaps, professional language service will help.
2. To show that UDP-glucose is the GCK degrader, the authors used UDP-Glucose to treat cells. However, whether endogenous UDP-glucose truly functions this way remains unclear. The UDP-glucose pyrophosphorylase (UGP) is the only enzyme that catalyzes the synthesis of UDP-Glucose from glucose-1-phosphate and UTP. The authors could deplete UGP by genetic approaches to examine whether GCK ubiquitylation and stability are altered.
Author Response
Dear Reviewer 2,
Please see the attachment.
Thank you.
Takeshi Imai

Round 2
Reviewer 2 Report
The revised version this manuscript is much improved.
I feel the paper is now acceptable for publication.